# Sustainability and the Digital Transition: A Literature Review

Albérico Travassos Rosário [1] and Joana Carmo Dias [2,*]

1 The Research Unit on Governance, Competitiveness and Public Policies (GOVCOPP), Universidade Europeia, 1200-649 Lisbon, Portugal; alberico@ua.pt

2 Centro de Investigação em Organizações, Mercados e Gestão Industrial (COMEGI), Universidade Lusíada, 1349-001 Lisbon, Portugal

* Correspondence: joana.carmo.dias@universidadeeuropeia.pt

**Abstract:** The digital transition processes have demonstrated an enormous capacity to develop and implement sustainable solutions, which allow solving several problems such as poverty, high rates of species extinction and lack of equal opportunity. However, little attention is paid to the connection between the digital transition and sustainability. Thus, a systematic bibliometric literature review was developed to fill this knowledge gap and demonstrate the potential contributions of the digital transition to environmental, economic, and social sustainability aspects. In environmental sustainability, the digital transition involves the application of technologies such as Artificial Intelligence (AI), big data analytics, Internet of Things (IoT), and mobile technologies that are used to develop and implement sustainability solutions in areas such as sustainable urban development, sustainable production, and pollution control. In economic sustainability, emerging digital technologies can drive transformation into a more sustainable circular economy, the digital sharing economy, and establish sustainable manufacturing and infrastructure design. In the digital transition to social sustainability, the studies analyzed demonstrate the need for multidimensional policy perspectives to address the current digital divide. For effective management of the digital transition that achieves sustainability goals, the study discusses alternative approaches that include innovation through experimentation and dynamic and sustainable advantages achievable through temporary benefits.

**Keywords:** sustainability; digital transition; sustainable development; systematic bibliometric literature review (LRSB)

## 1. Introduction

Despite the intensifying efforts in society, science, and technology to promote the co-existence of human civilization and Earth's biosphere, social and biophysical unsustainability indicators have been on the rise in recent years. Consequently, the concept of sustainability has received enormous attention from scholars and economists attempting to provide practical solutions. Most of the attention can be linked to the 2015 UN Summit, where the 2030 agenda for sustainable development was adopted. The World Commission on Environment and Development (WCED) defined it as "the development that meets the needs of the present without compromising the ability of future generations to meet their needs" [1] (p. 1948). The concept integrates multiple aspects of the current society, including ecological, social, and economic concerns. It also aims to balance environmental protection and economic growth. With the increased awareness of the need for sustainability, progress has been made, including increased use of renewable energy, strengthened protections for endangered species, and improved measures of environmental protection [2]. However, Fischer and Riechers [3] note that problems such as ongoing anthropogenic climate change, poverty, high rates of species extinction, and lack of equal opportunities continue to persist. These issues indicate the need to identify and develop advanced measures of measuring and solving unsustainability problems to protect the wellbeing of the current and future generations.

Consequently, scholars have identified emerging technologies as potential solutions. For example, Melnyk et al. [4] indicate that the third industrial revolution has formed "a green economy and harmonizes industrial metabolism with the metabolism of the biosphere" (p. 24). Emerging digital technologies have changed the way businesses and individuals engage in daily life and business activities, leading to a digital transition. The digital transition concept refers to the transition from analog to digital processes that allows digital tools to model processes and activities, thus improving performance and productivity. In addition, the digital transition has increased the capability to develop and implement sustainable solutions. For instance, the Artificial Intelligence (AI) alliance formed by Facebook, Amazon, Google, IBM, and Microsoft aims to create solutions for "humanity's most challenging problems, including making advances in health and wellbeing, transportation, education, and science" [5] (p. 60). In this case, the companies use AI technologies to create solutions to social concerns, which rank among the critical pillars of sustainability alongside environmental and economic concerns [6]. However, El Hilali et al. [5] indicate that minimal attention is paid to demonstrating the connection between digital transition and sustainability. Several research papers have been published to discuss the different points of view of sustainability and digital transition, but in an isolated way [7–10], yet very few papers have discussed the link between sustainability and digital transition. Consequently, the lack of research hinders practitioners from optimizing digital tools to provide sustainable solutions to the world's unsustainability issues. Therefore, this systematic bibliometric literature review aims to bridge the knowledge gap by synthesizing existing literature on sustainability and digital transition and enriching the published literature by discussing how to seize the opportunity of digital transition to solve sustainability problems.

## 2. Theoretical Background

Although the term 'sustainability' became mainstream in the 1980s, its origins can be tracked to many early roots. For example, Purvis et al. [6] note that in response to decreasing forest resources across Europe in the 17th and 18th centuries, Evelyn and Carlowitz introduced the concept of sustainable yield. In the 19th and 20th centuries, some ecologists and natural scientists advocated for preserving nature and the conservation of natural resources to ensure sustainable production and consumption. In addition, some political economists, including Smith, Ricardo, Mill, and Malthus, challenged the limits of population and economic growth and acknowledged the fundamental trade-offs between social justice and wealth accumulation [6]. Despite these activities, the sustainability concept did not gain global attention until the 20th century, when the Club of Rome's 'Limits to Growth' advocated for a sustainable world system and a sustainable society. In the 1960s, environmental movements began to critique practices and called for ecological conservation, leading to the 'sustainable development' concepts of the 1980s [11]. The current definitions of sustainability have been built on these concepts and expanded to accommodate the current environmental, social, and economic problems.

Consequently, sustainability can be defined from multiple perspectives, making it difficult to have a single definition of the concept. Eizaguirre et al. [12] indicated more than 100 definitions of sustainability used by practitioners and scholars to illustrate the concept based on varying contexts. However, the most globally accepted definition comes from the World Commission on Environment and Development, which defined sustainable development as growth that meets the needs and aspirations of the current populations without compromising those of future generations. Bellandi and De Propris [11] describe it as integrating economic vitality, environmental robustness, and social equity to develop resilient, healthy, diverse, and prosperous current and future communities. The operationalization of these definitions considers the economic, social, and environmental aspects of development. Consequently, Avila-Gutierrez et al. [13] argue that sustainable practices should ensure satisfactory outcomes for the environment and the global population while also promoting current and future generations' economic and social needs. Most scholars classify sustainability under three pillars, which can either be represented as the 3P's,

people, prosperity, and profits, or as the 3E's economy, environment, and equity [12]. The multiple aspects considered under these pillars include biodiversity, natural resources, sustainable urbanization, human rights, cultural diversity, distribution of resources, access to equal opportunities, security, and social cohesion, among others.

The rapid development of digital technologies has resulted in profound changes in operations and strategies in different sectors worldwide. For example, the automotive industry manufactures self-driving cars, computers' capability to recognize images has surpassed that of humans, and robots are used to automate manufacturing processes and phone calls in the customer care sector [14]. Consequently, companies have been prompted to improve their technologies, processes, and tools to embrace these developments and survive the digital disruption in a process termed as 'digital transition.' The digital transition involves automating some manual processes, improving turnaround times by adding additional integrations, and upgrading newer technologies [15]. While these processes guarantee improved efficiencies that translate to higher performance and productivity, they also pose a significant leadership challenge in keeping up with technological advancements. Fraga-Lamas et al. [16] argue that the primary leadership challenge created by these disruptive technologies involves transitioning the company towards a desired future position by frequenting, assessing, and revising its business strategies and roadmaps based on emerging intelligence and technologies. Thus, the digital transition is tied to innovation. It is a proactive approach that anticipates the next ample opportunity and implements appropriate measures to exploit it for the benefit of the organization and its key stakeholders.

The digital transition is not new since it has been an ongoing process involving adopting technologies as they emerge. The history of digital transition dates back to the 1950s, when companies began using digital technologies to facilitate operational and strategic change across different areas [14]. The client-server architecture, mainframe computers, and mini and personal computers allowed companies to centralize operations and decentralize responsibilities and related activities. Other innovations, such as mobile communications, smartphones, the internet, and cloud computing, enabled companies to create new business models and structures [15]. The transition in recent years has been geared towards newer technologies, including artificial intelligence (AI), internet of things (IoT), multi-cloud environments, big data, and distributed ledger technologies (DLT), which have significantly transformed business operations and strategies [16]. These innovations show that the transition process is an ongoing activity based on emerging technologies. Consequently, it is critical for companies and industries worldwide to develop clear roadmaps to strategize and implement the right approach to embrace and optimize these technologies.

While production practices are associated with improving the wellbeing of humankind, they have also been linked to the current environmental and industrial challenges. Consequently, scholars and practitioners have identified digital technologies as practical tools for transitioning the production processes towards sustainability [17]. Digital technologies can facilitate industrially sustainable through multiple ways, including establishing necessary change at the company level to enhance organizational performance towards sustainability, improving organizational planning processes to enable them to predict demand and identify opportunities presented by sustainability, and allowing the companies to experiment with new efficient business models [18]. The "construction industry and, in particular, 3D printing of concrete are profoundly changing construction technologies and construction processes. Materials engineering is still a challenge for the search for even more effective and performant 3D printable concrete" [19] (p. 1). Other examples of digital technologies used to achieve sustainability include sensor networks that improve the manufacturing processes' adaptability and flexibility. Despite identifying these benefits, their realization is not guaranteed [18]. Therefore, achieving them requires organizations to increase awareness and transformation of their manufacturing processes to achieve sustainability-related goals such as transitioning to a circular economy and optimizing materials and energy

consumption through high-performance machines, components, and robots. For example, the Industrial Internet of Things (IIoT) and data analytics can gather data from the design stage to the recycling stage that can be used to improve energy efficiency and product life cycle [20]. In addition, these digital technologies can be used to analyze data context to monitor performance and optimize productivity. Therefore, the digital transition involves a technological revolution that can improve industrial sustainability through intelligent management systems that can achieve multiple functions, including improving energy and resource efficiency and reducing waste.

One major aspect connecting the digital transition and sustainability is the increased demand for innovative sustainable ecosystems. Trading blocks and regions, countries, and clusters globally are experiencing ongoing structural changes and are trying to understand global technological shifts and innovation trends [17]. Consequently, implementing innovative sustainable ecosystems has become a critical technique for achieving and maintaining competitiveness in the global business environment. Costa and Matias [20] argue that innovation and problem-solving are strongly connected, with the former being used to provide solutions to complex problems, including those associated with sustainable development. Sustainable innovations can address long-term and short-term societal problems, promote cleaner production, and elevate domestic and international economic development goals [21]. For example, with sustainable technological developments, companies can initiate practices that promote human and environmental wellbeing and sustainable exploitation of resources. Therefore, sustainable innovations contribute to the development of solutions to societal problems by facilitating sustainable development through corporate sustainability, networks, and local communities.

Additionally, innovative sustainable ecosystems comprise a network of relationships involving various actors and objects. These key relationships establish connections that reinstate the significance of the environment and multiple institutions and facilitate the free flow of related information through value co-creation systems, leading to sustainability [17]. While interactive networks play a crucial role in generating and diffusing information and innovations, they require an environment that allows them to share the value with a society comprised of players with shared interests. Thus, the success of sustainable innovations depends on their acceptance among the broader communities worldwide. For example, the user community, governments, and players across the value chain should embrace the innovation ecosystem and communicate and promote it to facilitate the further digital transition through value creation. Costa and Matias [20] contribute to this argument by indicating that the sustainable innovation ecosystem can be reinforced by "enlarging participation to unusual partners" and ensuring flexibility and transparency (p. 3). Achieving sustainability requires resources and capabilities. Besides, the current environmental problems require appropriate advanced technological innovation, knowledge, and expertise [21]. The concept of innovative, sustainable ecosystems emphasizes collective intelligence as a primary way of accumulating the resources and capabilities required to achieve sustainability. In this case, the players across the network interact using their innovative mindset to establish and implement appropriate innovations that can solve the sustainability problems affecting the world.

## 3. Materials and Methods

One primary issue undermining sustainability efforts is the ambiguity, lack of clarity, and uncertainty associated with sustainability and digital transition. According to Salas-Zapata and Ortiz-Munoz [22], this problem "can hinder the operationalization of the concept [sustainability], generate contradictory discourses on the matter, and may affect the validity of the studies" (p. 153). A systematic bibliometric literature review is used in this research to provide new insights and analyze existing literature to eliminate this confusion and build knowledge on the connection between digital transition and sustainability. The justification for the choice of methodology is based on Dodgson's [23] argument that literature reviews are a form of research that uses a rigorous research process to gather valid

and reliable data needed to build knowledge. Xiao and Watson [24] further explain that literature analysis enhances readers' understanding of the breadth and depth of the existing literature and identifies gaps to explore. From this perspective, reviewing appropriate literature will enable identifying digital tools and opportunities that can be exploited to achieve sustainable goals and development.

In this sense, the systematic bibliometric literature review (LRSB) involves the screening and selection of information sources to ensure the validity and accuracy of the interpreted and presented data, the process was divided into 3 phases and 6 steps [25–27] (Table 1).

**Table 1.** Process of systematic LRSB.

| Fase | Step | Description |
|---|---|---|
| Exploration | Step 1 | Formulating the research problem |
| | Step 2 | Searching for appropriate literature |
| | Step 3 | Critical appraisal of the selected studies |
| | Step 4 | Data synthesis from individual sources |
| Interpretation | Step 5 | Reporting findings and recommendations |
| Communication | Step 6 | Presentation of the LRSB report |

Source: own elaboration.

The methodology approach began with a literature search on the SCOPUS indexing online database of scientific articles, the most important peer-reviewed peer in the academic world. However, we consider that the study has the limitation of considering only the SCOPUS database, excluding the other academic bases. The keyword "Sustainability" was used to identify potential sources during the initial search. A total of 282,887 documents were identified. Given the need to narrow down the references to the most relevant, the keyword "digital transition" was added. We decided not to include other keywords, such as digital transformation and digitalization, because, although similar, they are different concepts from digital transition, which may distort the object of study in this article. Inclusion criteria were limited to academic and scientific documents, including journal articles, books and book chapters, and conference articles. This step reduced the number of documents summarized in the final report to 36 (Table 2).

**Table 2.** Screening Methodology.

| Database Scopus | Screening | Publications |
|---|---|---|
| Meta-Search | Keyword: Sustainability | 282,887 |
| Inclusion Criteria | Keyword: Sustainability, Digital transition | 36 |
| Screening | Keyword: Sustainability, Digital transition Published until December 2021 | 36 |

Source: own elaboration.

Of the 36 scientific and/or academic documents, 24 are Articles; 6 Book Chapters; 4 Conference Papers; 2 Reviews.

## 4. Literature Analysis: Themes and Trends

Peer-reviewed documents on the topic until December 2021 were analyzed. The year 2021 was the year with the highest number of peer-reviewed documents on the subject, with 15 publications. Figure 1 analyzes peer-reviewed publications published through December 2021.

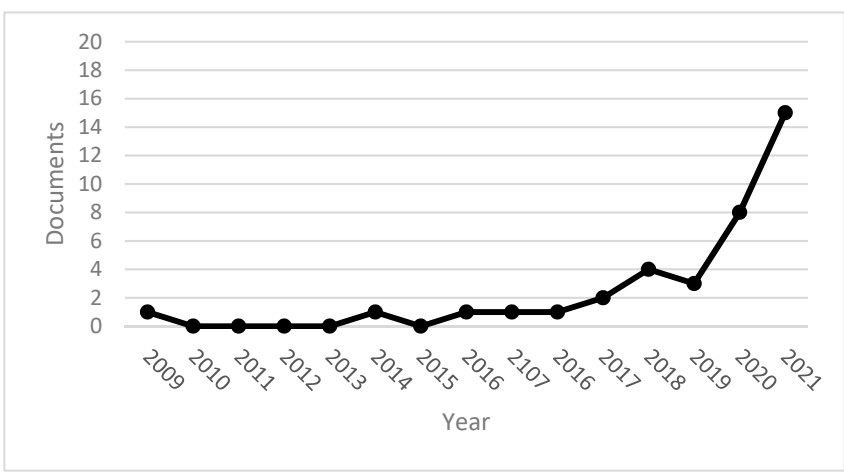

**Figure 1.** Documents by year. Source: own elaboration.

The publications were sorted out as follows: Sustainability Switzerland (10); Urban Book Series (3); Revesco Revista De Estudios Cooperativos (2); the remaining publications with a published scientific and/or academic document.

We can say that between 2019 and 2021 there has been an interest in research on Sustainability and the Digital transition.

In Table 3, we analyze the SCImago Journal & Country Rank (SJR), the best quartile and the H-index by publication. The *Journalism Studies* with 2.140 (SJR), Q1 and H-index 61.

There are a total of seven publications in Q1, five publications in Q2, one publication in Q3, and two publications in Q4. Publications from the best quartile Q1 represent 29% of the 24 publication titles; best quartile Q2 represents 21%, the best quartile Q3 represents 4%, and finally, the best quartile Q4 represents 8% of the 25 publication titles. Data from nine publications are not available.

As evident from Table 3, the significant majority of articles on Sustainability and the Digital transition rank on the Q1 best quartile index.

The subject areas covered by the 37 scientific articles were the following: Social Sciences (23); Energy (15); Environmental Science (13); Computer Science (8); Engineering (6); Economics, Econometrics and Finance (5); Business, Management and Accounting (3); Mathematics (3); Arts and Humanities (2); Chemistry (2); Decision Sciences (2); Materials Science (2); Physics and Astronomy (2); Biochemistry, Genetics and Molecular Biology (1); Earth and Planetary Sciences (1).

The most quoted article was "The core enabling technologies of big data analytics and context-aware computing for smart sustainable cities: a review and synthesis" from Bibri and Krogstie with 66 quotes published in the *Journal of Big Data* with 1.030 (SJR), the best quartile (Q1) and with an H-index (35). The study takes an approach to the new wave of computing based on smart, sustainable cities.

In Figure 2, we can analyze the evolution of citations of articles published between ≤2011 and December 2021. The number of citations shows a net positive growth with an R2 of 39% for the 2011–December 2021 period, with 2021 peaking at 87 citations.

The H-index was used to ascertain the productivity and impact of the published work, based on the largest number of articles included that had at least the same number of citations. Of the documents considered for the H-index, eight have been cited at least eight times.

In Appendix A, Table A1, the citations of all scientific and/or academic documents up to December 2021 are analyzed; 13 documents were not cited during this period, with a total of 198 citations. Appendix B, Table A2, examines the self-quotation of documents until 2021 of the 36 articles there were a total of 65 self-quotation "The core enabling technologies of big data analytics and con . . . "were self-cited 29 times.

**Table 3.** SCImago journal & country rank impact factor.

| Title | SJR | Best Quartile | H-Index |
|---|---|---|---|
| *Journalism Studies* | 2.140 | Q1 | 61 |
| *Journal of Cleaner Production* | 1.940 | Q1 | 200 |
| *Journal of Big Data* | 1.030 | Q1 | 35 |
| *International Journal of Energy Research* | 0.810 | Q1 | 95 |
| *International Journal of Sustainable Development and World Ecology* | 0.680 | Q1 | 43 |
| *Sustainability Switzerland* | 0.610 | Q1 | 85 |
| *Scires IT* | 0.450 | Q1 | 5 |
| *Sensors* | 0.640 | Q2 | 172 |
| *Energies* | 0.600 | Q2 | 93 |
| *Journal of Theoretical and Applied Electronic Commerce Research* | 0.560 | Q2 | 30 |
| *Revesco Revista De Estudios Cooperativos* | 0.510 | Q2 | 11 |
| *Public Finance Quarterly* | 0.400 | Q2 | 31 |
| *Aims Materials Science* | 0.370 | Q3 | 16 |
| *International Journal of Financial Studies* | 0.200 | Q4 | 6 |
| *Materiaux Et Techniques* | 0.180 | Q4 | 9 |
| *Ceur Workshop Proceedings* | 0.180 | - * | 52 |
| *Iop Conference Series Earth and Environmental Science* | 0.180 | - * | 26 |
| *Proceedings of the 3rd World Conference on Smart Trends in Systems Security and Sustainability Worlds4 2019* | 0.150 | - * | 4 |
| *Urban Book Series* | - * | - * | - * |
| *A Handbook of Digital Library Economics Operations Collections and Services* | - * | - * | - * |
| *East Asian Development Model Twenty First Century Perspectives* | - * | - * | - * |
| *Global Transitions* | - * | - * | - * |
| *Polito Springer Series* | - * | - * | - * |
| *Proceedings of the Design Society* | - * | - * | - * |

Note: * data not available. Source: own elaboration

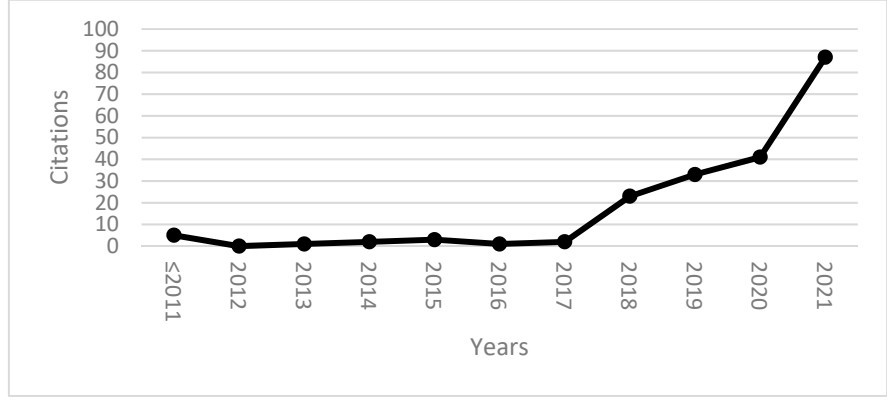

**Figure 2.** Evolution of citations between ≤2011 and 2021. Source: own elaboration.

In Figure 3, the bibliometric study is presented to investigate and identify indicators of the dynamics and evolution of scientific information. The study of bibliometric results using the scientific software VOSviewer 1.6.15 aims to identify the main research keywords in studies of Sustainability and Digital Transition.

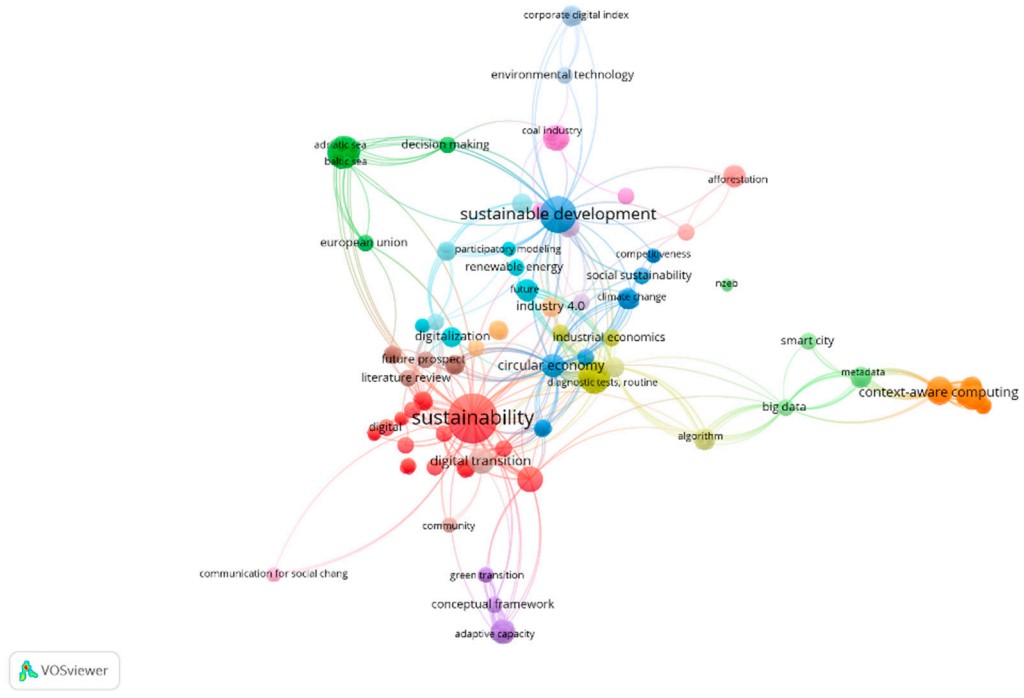

**Figure 3.** Network of all keywords.

The research was based on the articles analyzed on Sustainability and Digital Transition. The associated keywords can be examined in Figure 4, making clear the network of keywords that appear together/linked in each scientific article, thus allowing one to know the topics studied by the studies and identify future research trends. In Figure 5, a profusion of bibliographic couplings with a unit of analysis of cited references is presented.

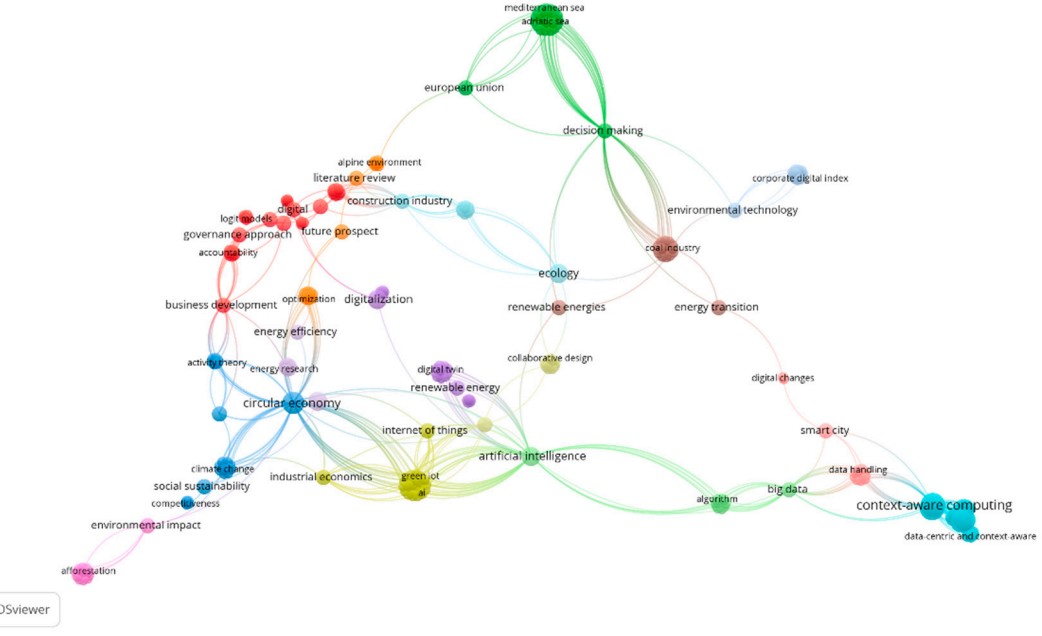

**Figure 4.** Network of Linked Keywords.

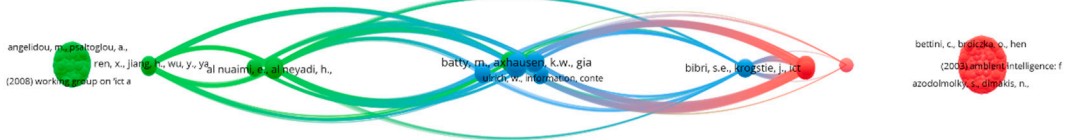

**Figure 5.** Networks bibliographic coupling.

## 5. Theoretical Perspectives

Sustainability catalyzes many organizations' embrace of digital technologies into all business fields. As the pressure for environmental responsibility increases, firms can adopt advanced technologies such as artificial intelligence (AI), machine learning (ML), predictive analytics, and the internet of things (IoT) to achieve sustainability goals [13]. Similarly, companies should integrate sustainability strategies into their digital transition roadmaps. For example, digital tools used in data tracking and sharing can identify and reduce environmental problems before they become magnified. Bellandi and De Propris [11] explain the interconnection between digital transitions and sustainability in the Fourth Industrial Revolution (FIR) by indicating that the real difference comes from the growing wave of combining multiple digital technologies with other transformations and innovations. For example, sustainability can be achieved and maximized by combining social and institutional innovations with nano- and bio-technologies (NBT), including renewable technologies, biogenetics, and neuro-technologies. Therefore, sustainability and digital transitions are interdependent, and changes in one affect the other.

### 5.1. Digital Transition for Environmental Sustainability

Digital transition can contribute to environmental sustainability. Technologies such as AI, big data analytics, IoT, social media, and mobile technologies are used to develop and implement sustainability solutions. For example, smart water management systems, an AI-based technology, are used to identify waterborne diseases in the waters [28]. Data collected from such technology can address the water pollution challenge and develop and implement solutions. In addition, AI, IoT, and big data analytics technologies enhance business practices' sustainability by reducing waste and carbon emissions [29]. Besides, these technologies have increased practitioners' capability to map the impact of various activities on the environment [30]. For example, big data analytics can assess direct environmental performance and enhance traceability within a specific system [21]. Thus, companies, governments, and non-governmental institutions can increase sustainability by embracing and integrating digital technologies into various activities, processes, and systems. Other sustainability aspects that can be improved with emerging technologies include sustainable urban development, waste management, sustainable production, and pollution control.

#### 5.1.1. Sustainable Urban Development

Urban sustainability refers to the idea of building cities without damage to the environment, with low waste production, and minimal use of natural resources. The primary categories of urban sustainability are smart cities and sustainable cities [31]. According to Feroz et al. [28], smart cities use new technologies such as IoT to improve people's lives, enhance working efficiency, and sustainably. Sustainable cities, on the contrary, enhance social wellbeing by using emerging technologies to control current resources in sustainable means [21]. Digital technologies such as big data and AI technologies foster

urban sustainability by providing unprecedented opportunities in sustainable and smart cities [29]. For instance, they can be used to lower carbon footprints in the city, creating a cleaner environment for human survival. Urban developers worldwide should combine appropriate technologies to ensure that the developed cities adhere to the current global sustainability goals.

### 5.1.2. Waste Management

Waste management involves various techniques applied to collecting, transporting, handling, and disposing of waste. In recent years, solid waste has become a critical environmental problem. As digital technologies gain global popularity, scholars and practitioners have recognized their capability to develop and implement practical solutions for waste management. Feroz et al. [28] identified the four most persistent types of waste, including solid waste, food waste, e-waste, and agri-waste. In Industry 4.0, companies have access to technologies that have significantly improved their capability to manage solid waste from industries such as iron and steel [32]. For instance, AI, IoT, and big data analytics can allow the identification of sustainability-related opportunities in solid waste management that prompt businesses to change their business models [33]. Similarly, big data can be used to identify illegal damping sites, while AI technologies collect and handle e-waste from electronic devices on demand from users. Thus, combining these digital technologies can help address complex waste management problems and drive sustainable innovations.

### 5.1.3. Sustainable Production

The digital transition has led to the adoption of cleaner and more sustainable production processes that reduce an organization's environmental impacts. Such a sustainable production system improves profitability, lowers operating costs, and increases employees' safety. The primary categories under sustainable production include smart manufacturing and sustainable supply chains that use various technologies, including IoT, cloud computing, cyber-physical systems, AI, digital twins, and big data analytics [28]. In the industry 4.0 era, the digital transition towards smart and sustainable manufacturing has gained popularity due to its advocacy for renewable energy consumption, sustainable manufacturing, and energy-saving. Consequently, the period is associated with achieving multiple sustainable goals, including reducing resource wastage, pollution, and environmental degradation, despite the progress made in achieving economic development [33]. Besides sustainable manufacturing, emerging technologies can transform logistics models throughout the supply chains and promote environmentally sustainable procurement procedures. The digital transition is an ongoing process of improving technologies that continues to strengthen sustainable manufacturing systems and methods.

### 5.1.4. Pollution Control

Rapid urbanization poses a threat to the environment and human health due to the high level of pollutants. Under pollution control, the digital transition can help address issues including $CO_2$ emissions, air pollution, climate change, disaster management, and water treatment. Feroz et al. [28] identify heavy industries such as iron and steel, chemical industries, and energy industries as the primary pollutants needing pollution control measures. Digital technologies provide advanced tools that improve how pollution is measured, controlled, and managed. For instance, AI is used for environmental pollution control proliferates, allowing practitioners to address the complex and dynamic pollution problems affecting the environment, and threatening human health [33]. In addition, emerging new technologies are being used to develop low-carbon transportation and green vehicles that will eventually lead to environmental sustainability by reducing carbon emissions. With innovations such as big data analytics and IoT sensors, companies have access to environmental data that can be used for sustainable decision-making. Therefore,

digital technologies increase the capability to measure and control pollution, eventually leading to ecological sustainability.

### 5.2. Digital Transition for Economic Sustainability

Digital technologies can play a significant role in accelerating progress towards achieving sustainable development and business growth. The increased use of digital technologies has revolutionized how institutional and economic systems conduct and organize tasks within uncertain and complex business environments. For instance, data technologies such as data science, big data, predictive analytics, smart metering, and forecasting play a significant role in facilitating data-driven development and business decisions [34]. Consequently, businesses are moving towards digital practices that can lead to economic sustainability through multiple changes in business models, demand for energy efficiency, and awareness of corporate responsibility in protecting the welfare of current and future generations [35]. Economic sustainability aspects discussed in this section include circular economy, digital sharing economy, and smart manufacturing, which are geared towards achieving sustainability.

### 5.2.1. Circular Economy

Digital transition can boost economic transformation towards a more sustainable circular economy (CE). CE is a business production and consumption model that involves sharing, renting, reusing, refurbishing, repairing, and recycling materials and products for the most extended duration possible. The use of digital technologies can help improve the CE model by building visibility, enhancing traceability, and transparency throughout the product's lifetime. García-Muiña et al. [36] argue that introducing digital technologies and connected objects can potentially decrease the use of resources and enhance circular systems. For instance, companies can use fewer resources more efficiently with appropriate digital technologies, thus reducing operational costs. In addition, digitization enables smart solutions that reduce energy consumption and facilitate the efficient use of capacity and logistics routes. Effective circular business models involve a network of connected players who capture and deliver value. Digital transition contributes to this performance and interconnection by providing digital tools and systems that increase transparency and enable the players to work together towards achieving common goals. For example, radio-frequency identification (RFID) technologies can be used to collect data on the use of a product and its movement from one consumer to another. This monitoring capability can help the key players ensure that the model is sustainable and does minimal harm to the environment and surrounding communities. Besides, by using data technologies, the key players can identify the specific regions where certain products are in demand and availing them for leasing and sharing to eliminate other unsustainable alternatives such as manufacturing and production that may be excessive and harmful to the environment.

### 5.2.2. Digital Sharing Economy

A digital sharing economy is an ICT-based resource allocation system performed by individuals and (non-) commercial organizations to allow sharing practices through digital platforms. The primary goal of the digital sharing economy is to facilitate access to material and immaterial resources for specific populations [37]. Social relationships develop significantly through these digital platforms, enabling the participants to engage in economic activities. Consequently, this technology-based economic activity is often referred to as "peer economy" due to the peer-to-peer type of transaction/networking involved. However, the sharing economy occurs in different forms, including peer-to-business (P2B), business-to-peer (B2P), and business-to-business (B2B) [37]. One major way that the sharing economy contributes to sustainability is by availing existing products on demand instead of producing new ones. Consequently, the economic process can reduce the environmental impacts associated with production processes [38]. In addition, since it optimizes purposeful social relations, the business model facilitates adequate communication flow that can be used to generate valuable sustainability data. For example, the peers engaged in

the sharing economy can freely share opinions on sustainable practices and implement practical solutions to enhance the economy's sustainability. Unlike corporate-controlled economies, a peer-to-peer economy does not involve the complex bureaucracies that delay decision-making and the implementation of sustainable business practices.

The sharing economy challenges the old monopolies and creates a socially connected economy. A change in ownership of the products within a social network eliminates the need to buy new products. Consequently, the economic structure reduces waste by ensuring that the products are used until they wear off instead of throwing them away once an individual no longer needs them [37]. For example, an individual can choose to sell clothes that no longer fit at a lower price to other potential buyers instead of throwing them away when they are still in usable condition. In this case, individuals engage in sustainable business practices without engaging the formal business sector that was traditionally the predominant product and service provider [38]. The sharing economy proves that achieving desired sustainability levels requires the involvement of both corporates and individuals. Emerging technologies such as smartphones and the internet simplify this process by providing networking channels that connect people who conduct digital exchanges.

### 5.2.3. Sustainable Manufacturing and Infrastructure

The life cycle environmental impacts and depletion of resources will severely undermine the design and manufacturing of products for future generations. Thus, sustainable manufacturing and infrastructure have become a critical way of ensuring that current business practices do not compromise those of future challenges. In this regard, sustainable manufacturing can be defined as a systematic approach of creating and distributing innovative products and services in a way that eliminates excessive use of resources such as water, land, and energy, produces zero waste to reduce $CO_2$ emissions, and eradicates toxic substances [39]. Digital transition plays a critical role in availing appropriate technologies needed to manufacture products in sustainable ways that align with the globally accepted sustainable development goals that involve meeting current generations' goals and needs without compromising future generations [40]. For example, under sustainable manufacturing, companies are implementing new analysis procedures and designs to reduce environmental impacts. In addition, they use technologies such as AI-powered robots and machines to improve material handling practices and reduce energy and material consumption. Therefore, sustainability has become a core element in modern-day manufacturing processes.

### *5.3. Digital Transition for Social Sustainability*

Social sustainability refers to all those formal and informal processes, relationships, systems, and structures employed to ensure there is adequate support for healthy living by the current and future generations. It involves identifying and managing positive and negative business impacts on communities and people [41]. The new technological developments and the digital transition technologies that have spread in all areas of life have caused considerable social effects [42]. This is because the digital transition impacts and fundamentally changes individuals' habits. In turn, such practices are likely to affect future generations' lifestyles and quality of life. The continued digital society's emergence implies that social sustainability has become critical in the digital transition. Scientific studies have clearly expressed the existing positive association between digital transition and social sustainability. Globally, sustainable development agendas are based on inclusiveness and shared prosperity ideals. Social justice principles form a critical component as nations work to address interlink between the dimensions of sustainable development, including social, economic, and environmental issues.

One of the significant issues is that the current digital inequality reinforces the existing social inequalities. Although the digital divide cannot be wholly eliminated, significant gains can be made if access to digital media such as the internet can be improved for the whole global population. This should also include reducing inequalities in digital skills and usage. Apart from physical access, which has been the primary focus of the digital

divide policies, other areas becoming more important include building digital skills and internet usage opportunities [43]. Studies also identify the need for multidimensional policy perspectives to solve the current digital divide. This implies that such policies should be persuasive or work to create awareness and focus on technological, educational, economic, and social perspectives [44]. To effectively achieve digital transition for social sustainability goals, a digital technology perspective for social development and wellbeing should target the following three areas: employment and job markets, education, and healthcare.

### 5.4. Managing Digital Transition to Facilitate Sustainability

Managing digital transition requires organizational leadership to make bold decisions and implement actions to help achieve a sustainable competitive advantage. Digital transitions that would help facilitate sustainability are associated with considerable complexities and risks, forcing organizational leadership to make one big decision and hope for the best outcomes [45]. Three new approaches have been identified as practical alternatives to the traditional linear and big bang approaches when considering sustainable digital transition strategies. The new alternative approaches include innovating by experimenting, incremental approaches, and dynamic, sustainable advantages achievable through temporary advantages.

### 5.4.1. Innovating by Experimenting

Many organizations have maintained the traditional annual or multi-year cycles and execution strategies despite growing uncertainties in the modern business environment. In a business environment characterized by an uncertain future and continuously shifting destinations and paths, business leaders are essentially required to continually and regularly evaluate and update strategic plans [46]. Therefore, iterative and learning processes are necessary to formulate strategies and execute and implement actions to recalibrate such a strategy. Innovating by experimenting is one of the popular approaches that also enables a business to inexpensively test numerous new ideas while at the same time considering sustainability. Both the internal and external sources can provide emerging intelligence that would help to evaluate such new ideas. In addition, this strategy requires business leaders to scale up a working idea rapidly. In cases where a new idea does not work, business leaders should move on to other ideas before making significant losses. Compared to the traditional approaches, innovating by experimenting is more effective since it allows organizational leadership to test and learn from new ideas. Corporations such as Alibaba, Google, Amazon, and Didi Chuxing have used this approach to achieve critical success. Overall, digital technologies can play a crucial role in enabling new strategies and operations that deliver excellent results in transitions that consider sustainability.

### 5.4.2. Incremental Approaches for Radical Transformations

The modern digital economy is significantly different from the service or industrial economies in several ways. The markets currently comprise notable changes in the game's rules and the key players. These changes create a significant mismatch between the digital future and existing traditional business models [47]. Many organizations identify such a mismatch as a too big to bridge gap in the business environment. However, a radical transformation that considers sustainability issues does not have to involve one big step in the planning and implementation phases [48]. Relatively, a series of incremental steps can be adopted to achieve the much-desired radical transformation. An example is using an outcome-driven approach by some prominent business organizations to ensure that desired results are delivered at each stage of digital transition initiatives. In incremental systems, rapid piloting and scaling become the basis for experimenting with several new ideas. The proposed approach involves splitting the large-scale radical digital transition into smaller, more manageable strategic investments. The process allows businesses to test and nurture innovations while at the same time avoiding potential risks and achieving desired sustainability goals [49]. Therefore, the approach involves a series of incremental steps to achieve digital transition while effectively mitigating high risks. The method

differs from the traditional big bang approach since it requires corporate leadership to consider initial up-front investment, sustainability, and changes to the balance before making investment decisions.

### 5.4.3. Evolving Portfolio of Temporary Advantages

The evolving portfolio of temporary advantages will play a key role in achieving dynamic, sustainable advantages. Although the digital transition's key objective is to achieve sustainable competitive advantages (SCAs), only a few competitive advantages are genuinely sustainable for prolonged periods in the digital economy. Developments such as innovations and competitors' imitations rapidly erode the competitive advantages of the digital economy, making them temporary or transient [50]. Continuously experimenting with an evolving portfolio comprising incremental and radical innovations is helping corporate leadership pursue successive temporary advantages. Although they achieve small gains from each temporary benefit, the cumulative effects in the long term are significant [51]. The introduction of successive temporary advantages before the erosion of older ones allows for the dynamic achievement of SCAs in the evolving temporary advantage portfolio. The approach presents additional benefits, such as ensuring that strategy remains as a direction for action and not merely a predefined plan. Therefore, the strategy encourages organizational leaders to focus on formulating and implementing short-term decisions while at the same time considering the long-term strategy and destination [52]. Business leaders can use this approach to explore alternative routes frequently and, in some cases, change their destination, intertwine the strategy and execution, and use emerging intelligence to inform sustainable evaluation and recalibration of strategic directions.

## 6. Conclusions

This study is a literature review of the recent developments in sustainability and digital transition. The review is needed because of the recent increase in social and biophysical unsustainability indicators recorded in recent years, despite continued efforts by science and technology to promote the co-existence of human civilization and the Earth's biosphere. The digital transition has been identified as a disruption period that began in the 1950s that involved automating some manual processes, improving turnaround times by adding additional integrations, and upgrading to newer technologies. Sustainability is a concept that has been associated with environmental, social, and economic problems for which emerging digital technologies could provide practical solutions. In sustainability, the economic, environmental, and social aspects are seen as essential pillars for ensuring present human needs are achieved without compromising the ability of future generations to meet their needs.

The existing studies demonstrate the potential contributions of the digital transition to environmental, economic, and social sustainability aspects. In environmental sustainability, the digital transition involves the application of technologies such as AI, big data analytics, IoT, social media, and mobile technologies that are used to develop and implement sustainability solutions in areas such as sustainable urban development, waste management, sustainable production, and pollution control. In economic sustainability, emerging digital technologies can boost transformation in the more sustainable circular economy (CE), the digital sharing economy, and establish sustainable manufacturing and infrastructure design. In the digital transition for social sustainability, the reviewed studies demonstrate a need for multidimensional policy perspectives to solve the current digital divide. These policies should reduce the existing divide in access, skills, and usage of existing digital technologies. The studies demonstrate that digital technology perspectives for social development and wellbeing should target the following three areas: employment and job markets, education, and healthcare. For effective management of the digital transition that achieves sustainability goals, the study identifies and discusses new alternative approaches that include innovating by experimenting, incremental strategies, and dynamic, sustainable advantages achievable through temporary benefits. Moreover, this research has

made a theoretical contribution by highlighting digital technologies as practical tools for transitioning the production processes towards sustainability.

**Author Contributions:** Conceptualization, A.T.R. and J.C.D.; methodology, A.T.R. and J.C.D.; software, A.T.R. and J.C.D.; validation, A.T.R. and J.C.D.; formal analysis, A.T.R. and J.C.D.; investigation, A.T.R. and J.C.D.; resources, A.T.R. and J.C.D.; data curation, A.T.R. and J.C.D.; writing—original draft preparation, A.T.R. and J.C.D.; writing—review and editing, A.T.R. and J.C.D.; visualization, A.T.R. and J.C.D.; supervision, A.T.R. and J.C.D.; project administration, A.T.R. and J.C.D.; funding acquisition, A.T.R. and J.C.D. All authors have read and agreed to the published version of the manuscript.

**Funding:** This research is supported by national funding's of FCT—Fundação para a Ciência e a Tecnologia, I.P., in the project «UIDB/04005/2020».

**Institutional Review Board Statement:** Not applicable.

**Informed Consent Statement:** Not applicable.

**Data Availability Statement:** Not applicable.

**Acknowledgments:** We would like to express our gratitude to the editor and the referees. They offered valuable suggestions or improvements. The authors were supported by the GOVCOPP Research Center of the University of Aveiro, and COMEGI.

**Conflicts of Interest:** The authors declare no conflict of interest. The funders had no role in the design of the study; in the collection, analyses, or interpretation of data; in the writing of the manuscript, or in the decision to publish the results.

## Appendix A

**Table A1.** Overview of document citations period ≤2011 to 2021.

| Title | Date | ≤2011 | 2012 | 2013 | 2014 | 2015 | 2016 | 2017 | 2018 | 2019 | 2020 | 2021 | Total |
|---|---|---|---|---|---|---|---|---|---|---|---|---|---|
| Green IOT and Edge AI as Key Technological Enablers for a Su . . . | 2021 | - | - | - | - | - | - | - | - | - | - | 1 | 1 |
| Definition of the Future Skills Needs of Job Profiles in the . . . | 2021 | - | - | - | - | - | - | - | - | - | - | 5 | 5 |
| Digitalization as a Strategic Means of Achieving Sustainable . . . | 2021 | - | - | - | - | - | - | - | - | - | - | 4 | 4 |
| Small and Medium-Sized Ports in the Ten-t Network and Nexus . . . | 2021 | - | - | - | - | - | - | - | - | - | - | 3 | 3 |
| Smart Villages Policies: Past, Present and Future | 2021 | - | - | - | - | - | - | - | - | - | - | 3 | 3 |
| A Win-Win case of CSR 3.0 for Wellbeing Economics: Digital . . . | 2021 | - | - | - | - | - | - | - | - | - | - | 3 | 3 |
| Benefits of BIM Implementation in the French Construction in . . . | 2020 | - | - | - | - | - | - | - | - | - | - | - | 1 |
| Open Innovation 4.0 as an Enhancer of Sustainable Innovation . . . | 2020 | - | - | - | - | - | - | - | - | - | 1 | 23 | 24 |
| Eco-Holonic 4.0 Circular Business Model to Conceptualize Sus . . . | 2020 | - | - | - | - | - | - | - | - | - | - | 5 | 5 |
| Competitiveness as a Decisive Criterion for Sustainability | 2020 | - | - | - | - | - | - | - | - | - | 1 | 6 | 6 |
| Participatory Development of Digital Support Tools for Local . . . | 2020 | - | - | - | - | - | - | - | - | - | - | 3 | 3 |
| The Role of 'Digitalization' in German Sustainability Bank r . . . | 2020 | - | - | - | - | - | - | - | - | - | - | 1 | 1 |

**Table A1.** *Cont.*

| Title | Date | ≤2011 | 2012 | 2013 | 2014 | 2015 | 2016 | 2017 | 2018 | 2019 | 2020 | 2021 | Total |
|---|---|---|---|---|---|---|---|---|---|---|---|---|---|
| The Role of a Digital Industry 4.0 in a Renewable Energy Sys . . . | 2019 | - | - | - | - | - | - | - | - | 3 | 4 | 9 | 16 |
| Digital Transitions: The Evolving Corporate Frameworks of le . . . | 2019 | - | - | - | - | - | - | - | - | 2 | 3 | 3 | 8 |
| Rationalizing a Personalized Conceptualization for the Digit . . . | 2018 | - | - | - | - | - | - | - | 1 | 5 | 3 | 3 | 12 |
| Sustainable Urban Forms: Time to Smarten up with Big Data An . . . | 2018 | - | - | - | - | - | - | - | - | - | - | 1 | 1 |
| Big Data Analytics and Context-Aware Computing: Core Enablin . . . | 2018 | - | - | - | - | - | - | - | - | - | - | - | 1 |
| The Core Enabling Technologies of Big Data Analytics and Con . . . | 2017 | - | - | - | - | - | - | - | 11 | 22 | 23 | 10 | 66 |
| Unintended Side Effects of Digital Transition: Perspectives . . . | 2017 | - | - | - | - | - | - | - | 2 | - | 4 | 2 | 8 |
| Big Data and Context-Aware Computing Applications for Smart . . . | 2016 | - | - | - | - | - | - | - | 6 | 1 | 1 | - | 8 |
| Introduction: The East Asian Development model | 2014 | - | - | - | - | - | - | 1 | - | - | - | - | 1 |
| Roles of Digital Technology in China's Sustainable Forestry . . . | 2009 | 5 | | 1 | 2 | 3 | 1 | 1 | 3 | - | - | 2 | 18 |
| | Total | 5 | 0 | 1 | 2 | 3 | 1 | 2 | 23 | 33 | 40 | 88 | 198 |

Source: original source.

## Appendix B

**Table A2.** Overview of document self-citation period ≤2011 to 2021.

| Title | Date | ≤2011 | 2012 | 2013 | 2014 | 2015 | 2016 | 2017 | 2018 | 2019 | 2020 | 2021 | Total |
|---|---|---|---|---|---|---|---|---|---|---|---|---|---|
| Small and Medium-Sized Ports in the Ten-t Network and Nexus . . . | 2021 | - | - | - | - | - | - | - | - | - | - | 2 | 2 |
| Smart Villages Policies: Past, Present and Future | 2021 | - | - | - | - | - | - | - | - | - | - | 2 | 2 |
| Open Innovation 4.0 as an Enhancer of Sustainable Innovation . . . | 2020 | - | - | - | - | - | - | - | - | - | - | 2 | 2 |
| Eco-holonic 4.0 Circular Business Model to Conceptualize Sus . . . | 2020 | - | - | - | - | - | - | - | - | - | - | 1 | 1 |
| The Role of a Digital Industry 4.0 in a Renewable Energy Sys . . . | 2019 | - | - | - | - | - | - | - | - | - | - | - | 1 |
| Rationalizing a Personalized Conceptualization for the Digit . . . | 2018 | - | - | - | - | - | - | - | - | 5 | 3 | 3 | 11 |
| The Core Enabling Technologies of Big Data Analytics and Con . . . | 2017 | - | - | - | - | - | - | - | 6 | 11 | 11 | 1 | 29 |
| Unintended Side Effects of Digital Transition: Perspectives . . . | 2017 | - | - | - | - | - | - | - | 2 | - | 1 | - | 3 |
| Big Data and Context-Aware Computing Applications for Smart . . . | 2016 | - | - | - | - | - | - | - | 6 | - | - | - | 6 |
| Roles of Digital Technology in China's Sustainable Forestry . . . | 2009 | 4 | - | 1 | 3 | - | - | - | - | - | - | - | 8 |
| | Total | 4 | 0 | 1 | 3 | 0 | 0 | 0 | 14 | 16 | 15 | 11 | 65 |

Source: original source.

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
