# Peer review of "Sustainability and the Digital Transition: A Literature Review"

_sustainability, doi:10.3390/su14074072_

Round 1

Reviewer 1 Report

Dear Authors
You consider that this is an article on a current topic of interest for the journal. 
I would like to know why only the scopus database has been used.I would also consider it appropriate to increase the keywords.
Line 95 talks about exclusion criteria and table 2 about inclusion criteria. You should clearly specify the inclusion and exclusion criteria.

Author Response

(1) Dear Authors I consider that this is an article on a current topic of interest for the journal.I would like to know why only the scopus database has been used.

Thanks for the comment. The use of Scopus alone is due to the fact that it is the main article base for academic journals/magazines, covering around 19,500 titles from more than 5,000 international publishers, including coverage of 16,500 peer-reviewed journals in the fields scientific, technical, and medical and social sciences. Thus providing a very real view of the researched  subjects with scientific and/or academic relevance.

(2) I would also consider it appropriate to increase the keywords.

The authors thank the comment and suggestion. We've added more keywords such as sustainable development, and systematic review of bibliometric literature (LRSB).

(3) Line 95 talks about exclusion criteria and table 2 about inclusion criteria. You should clearly specify the inclusion and exclusion criteria.

Thank you for bringing this to our attention. The text on line 95 has been clarified. Our inclusion criteria were academic and scientific documents, including journal articles, books and book chapters, and conference articles.

Reviewer 2 Report

Major changes required:
(1) Describe (in the Introduction) the real research gap addressed by the paper by providing recent contributions which highlight and document such gap;
(2) Include a section 2 on theory background, i.e. literature representing the state-of-art knowledge on the topic and which presents the current contributions which the authors aim to advance in relation to the identfied research gap;
(3) the method is not clearly described. How "keywords" were defined? Why those keywords and not others? Why "digital transitions" and not "digital transformation" or "digitalization" or many other possible? Moving from 282,887 to 36 papers is really hard to obtain and it is doubtful that in 2022 there are only 36 research articles providing conceptual frameworks or practical experiences on digital/sustainability; the authors should be much more specific on how keywords are chosen, the inclusion/exclusion criteria and the gradual process (not from 282,887 to 36!) of selecting the papers to be analysed;
(4) It is really not clear what the section 4 (Theoretical Perspectives) represents; if theory background, then it should be used in a new section  (2, on background); if outcomes of the SLR, then it should organized more effectively: why those (many) sub-sections? How they were obtained moving from the SLR? what is the value added provided by the authors? since this looks the main contribution of the paper, how this section 4 provides an advancement respect to the identified gap?
(5) Include a last section with real theory implications (advancements respect to extant knowledge) and practitioner implications (the topic has many practical views);
(6) Have the paper be proofread by English mothertongue.

Author Response

(1) Describe (in the Introduction) the real research gap addressed by the paper by providing recent contributions which highlight and document such gap.

Yes, you are right. So, in the Introduction section with added the research gap addressed by this paper - the study of the digital transition as a lever to solve the sustainability problems that still persist.

(2) Include a section 2 on theory background, i.e. literature representing the state-of-art knowledge on the topic and which presents the current contributions which the authors aim to advance in relation to the identified research gap.

The authors thank the reviewer for valuable suggestion to strengthen the manuscript quality. We added the section Theoretical Background, after the Introduction, in order to present the state-of-art about sustainability and digital transition

(3) The method is not clearly described. How "keywords" were defined? Why those keywords and not others? Why "digital transitions" and not "digital transformation" or "digitalization" or many other possible? Moving from 282,887 to 36 papers is really hard to obtain and it is doubtful that in 2022 there are only 36 research articles providing conceptual frameworks or practical experiences on digital/sustainability; the authors should be much more specific on how keywords are chosen, the inclusion/exclusion criteria and the gradual process (not from 282,887 to 36!) of selecting the papers to be analysed.

The authors thank the reviewer for the suggestion. The keywords mentioned are in line with the research “Sustainability and the digital transition: a literature review”. The non-use of other keywords such as "digital transformation" or "digitalization" is related to the fact that we consider that they are different concepts, although very closed ones, and will certainly give different studies. In terms of the existence of only 36 documents, we don’t have a elaborated explanation as the tracking output is done in the Scopus indexing database. With the simple step of including the keywords "sustainability" and "digital transition" and selecting academic and scientific documents, such as journal articles, books and book chapters, and conference articles, we obtained 36 results.

(4) It is really not clear what the section 4 (Theoretical Perspectives) represents; if theory background, then it should be used in a new section (2, on background); if outcomes of the SLR, then it should organized more effectively: why those (many) sub-sections? How they were obtained moving from the SLR? what is the value added provided by the authors? since this looks the main contribution of the paper, how this section 4 provides an advancement respect to the identified gap?

Thank you for calling our attention to this issue. In fact, we agree with the lack of clarity. The section Theoretical Perspectives represents the outcomes of the SLR and

consists of the most relevant and discussed topics in the literature. This section presents the potential opportunities for collaboration between digital transition and sustainability.

However, the first part of this section serves as theoretical background so we have reorganized this section to make it clearer, and opened a Theoretical Background section after the Introduction.

Reviewer 3 Report

This version of the manuscript was already reviewed by other reviewers; thus the authors have revised. All in all, the format is good, and the source of the prior research articles are basically selected form SCI (SSCI) journal, which makes their sources with certain quality. The figures of network of all keywords are pretty impressive, which makes the connections between articles clearer. There is no further revision needs to be carried out. But please re-check the spelling and grammar before it published.

Reviewer 4 Report

 The authors successfully responded to the reviewer's comments and adapted the scientific paper in accordance with those comments. 

Round 2

Reviewer 2 Report

Unfortunately, the authors missed to address the most important criticism reported in the first round of reviews, i.e. the poor approach to conducting the SLR. The use of alternative wording is crucial in a paper search and selection process.

Again, how were keywords defined? Why "digital transition", "digital transformation", "digitalization" and many other possible are not considered as search terms? Moving from 282,887 to 36 papers is hard (not credible) to justify in a scientifically robust manner.

The authors respond that "keywords mentioned are in line with the research “Sustainability and the digital transition: a literature review”".

Well, keywords should not be "in line" with the title but rather with the  research field and questions (which are not necessarily restricted/contained in the title).

Still, the authors argue that "The non-use of other keywords such as "digital transformation" or "digitalization" is related to the fact that we consider that they are different concepts, although very closed ones, and will certainly give different studies."

Of course, "digital transition" OR "digital transformation" OR "digitalization" OR many other should be used in searches COMBINED with "sustainability". For example, when searching "digital transformation"  AND "sustainability", one can obtain 103 results in Scholar and it is not clear what is the level of overlapping (if any) with the search criteria defined by the authors.

As it stands now, the literature review is not scientifically valid and comprehensive; all the considerations and conclusions matured by the authors based on the SLR cannot thus be assuemed as valid or verified.

Author Response

We greatly appreciate the comment made by the reviewer and we consider that the suggested keywords are important suggestions for future research. For this particular article, we fear that it is not a viable solution, as we tried to follow the reviewer's suggestions and obtained a total of 316,296 scientific documents through the following research stream:

TITLE-ABS-KEY ( "Sustainability") OR TITLE-ABS-KEY ( "Digital transition" ) OR TITLE-ABS-KEY ( "digital transitions" ) OR TITLE-ABS-KEY ( "digital transformation" ) OR TITLE- ABS-KEY ("digitalization")

We are faced with a high number of documents that make their reading and consequent literature review impractical.

The reviewer mentions that he tested the search for "digital transformation" AND "sustainability" on Scholar and obtained 103 results. However, what we intend is to use a very clear and rigorous scientific methodology that makes this research procedure credible and replicable. Therefore, we use a scientific database with indexed documents of scientific relevance such as SCOPUS.

We also understand that moving from 282,887 to 36 papers could be hard to justify in a scientifically robust manner. However, it only results from the methodological procedure mentioned above that makes the literature review valid.

Also, although the suggested keywords are very important and that we would like to use in future research, they constitute different concepts from the keywords selected in the first place, which may eventually move away the results of the article from our object of study. We do not intend that the keywords are in line with the title of the article but that they are in line with our object of study which is to bridge the knowledge gap by synthesizing existing literature on sustainability and digital transition and to enrich the published literature by discussing how to seize the opportunity of digital transition to solve sustainability problems. And because we feel the need to add this justification in the Materials and Methods section, we add the following sentence “We decided not to include other keywords, such as digital transformation and digitalization, because, although similar, they are different concepts from digital transition, which may distort the object of study of this article.” (underlined in blue in the article).

Kind regards,

Round 3

Reviewer 2 Report

The Scholar search was just an example. Of course, the SLR has to based on Scopus and WoS queries. The explanation of the authors is not convincing. As it stands now, the method is weak and it is not possible to extract robust or scientifically valid considerations. Digital transformation and digitalization cannot be considered as external concepts respect to the authors' field of investigation.
If not in the abstract (which may drive too much results) at least in the title a larger search has to be performed. One example is the following:

TITLE ( "Sustainability" )  OR  TITLE ( "Sustainable" )  AND  TITLE ( "digital transition" )  OR  TITLE ( "digital transformation" )  OR  TITLE ( "digitalization" ) 

The authors are thus invited to expand their  search in order to address a broader population of research articles, which may finally support the generalizations and outcomes expected by the authors.

Author Response

Dear reviewer,

We appreciate the improvement suggestion and fully agree with your position. However, we will decline your suggestion since, on the one hand, we would have to do a literature review from the beginning and, on the other hand, the number of articles makes this literature review unfeasible.

Sincerely,